# Global sounding of F region irregularities by COSMIC during a geomagnetic storm

Klemens Hocke[1,2], Huixin Liu[3], Nicholas Pedatella[4], and Guanyi Ma[5]

[1]Institute of Applied Physics, University of Bern, Bern, Switzerland
[2]Oeschger Centre for Climate Change Research, University of Bern, Bern, Switzerland
[3]Kyushu University, Fukuoka, Japan
[4]High Altitude Observatory, National Center for Atmospheric Research, Boulder, Colorado, USA
[5]National Astronomical Observatories, Chinese Academy of Sciences, Beijing, China

*Correspondence to:* K. Hocke
(klemens.hocke@iap.unibe.ch)

**Abstract.** We analyze reprocessed electron density profiles and TEC profiles of the ionosphere in September 2008 (around solar minimum) and September 2013 (around solar maximum) obtained by the Constellation Observing System for Meteorology, Ionosphere, and Climate (COSMIC/FORMOSAT-3). The TEC profiles describe the total electron content along the ray path from the GPS satellite to the low Earth orbit as function of the tangent point of the ray. Some of the profiles in the magnetic polar regions show small-scale fluctuations with spatial scales $< 50$km. Possibly the trajectory of the tangent point intersects spatial electron density irregularities in the magnetic polar region. For derivation of the morphology of the electron density and TEC fluctuations, a 50 km high pass filter is applied in the s-domain where s is the distance between a reference point (bottom tangent point) and the tangent point. For each profile, the mean of the fluctuations is calculated for tangent point altitudes between 400 and 500 km. First at all, the global maps of $\Delta N_e$ and $\Delta$TEC are quite similar. However, $\Delta$TEC might be more reliable since it is based on less retrieval assumptions. We find a significant difference if the arithmetic mean or the median is applied to the global map of September 2013. In agreement with literature, $\Delta$TEC is enhanced during the post-sunset rise of the equatorial ionosphere in September 2013 which is associated with spread $F$ and equatorial plasma bubbles. The global map of $\Delta$TEC at solar maximum (September 2013) has stronger fluctuations than those at solar minimum (September 2008). We obtained new results when we compare the global maps of the quiet phase and the storm phase of the geomagnetic storm of 15 July 2012. It is evident that the TEC fluctuations are increased and extended over the Southern magnetic polar region at the day of the geomagnetic storm. The North-South asymmetry of the storm response is more pronounced in the upper ionosphere (ray tangent points h=400-500km) than in the lower ionosphere (ray tangent points h=200-300km).

## 1 Introduction

GPS radio occultation can be regarded as a bistatic limb sounding of the atmosphere where the transmitter is on a GPS satellite and the receiver is on a Low Earth Orbit (LEO) satellite. The technique was described in detail by Rocken et al. (1997) and

Hajj et al. (2002). Since the GPS radio occultation technique performs atmospheric limb sounding, the vertical resolution is about 1 km or better in the troposphere.

GPS radio occultation was already utilized to derive global maps of sporadic $E$ layers around 90-120 km altitude (Arras et al., 2010; Wu et al., 2005; Hocke et al., 2001). Our study aims at the retrieval of global maps of the amplitude of small-scale ionospheric irregularities with scales from 2-50 km in the ionospheric $F_2$ region between 400 and 500 km altitude using the GPS radio occultation technique. $F$ region irregularities are inducing phase and amplitude scintillations in radio signals. Aarons (1982) showed a scheme of a global scintillation map where the scintillations are strong at high geomagnetic latitudes in the polar caps or after sunset around the geomagnetic equator (20°S to 20°N). Fejer and Kelley (1980) classified the $F$ region irregularities into three groups: equatorial spread $F$, high-latitude irregularities and mid-latitude irregularities. Satellite measurements of the electric field and plasma density fluctuations indicate that the high-latitude ionosphere is highly structured and bounded by an extremely sharp transition between the disturbed polar region and the quiet ionosphere outside (Fejer and Kelley, 1980). The authors identify three main sources in production of high-latitude irregularities. There are production by particle precipitation, generation by electrostatic turbulence, and plasma instabilities. In addition, quasi-direct current electric fields play an essential role in transporting irregular plasma at high-latitudes (Fejer and Kelley, 1980).

Recently, global distributions of topside ionospheric irregularities (above the LEO orbit) were retrieved by using in situ data of LEO satellites (Zakharenkova and Astafyeva, 2015). Hocke et al. (2002) attempted to extract the fluctuations of total electron content along the GPS-LEO link at tangent point altitudes between 400 and 600 km. However, the global coverage of the occultation events of the early GPS/MET experiment was poor, and the measurement phase was during solar minimum in 1995 when less ionospheric irregularities are expected. Another approach is the use of the ground station network of GPS and GLONASS receivers. Cherniak and Zakharenkova (2017) monitored high-latitude ionospheric irregularities during the geomagnetic storm of June 2015 and derived polar maps of the distribution of the plasma irregularities based on the observations of the ground station network. Carter et al. (2013) analysed the scintillations of the GPS signals received by COSMIC-FORMOSAT-3. They found a spatio-temporal distribution of the GPS scintillations which is similar to those of equatorial spread $F$ and equatorial plasma bubbles. However, high-latitude $F$ region irregularities were not found by Carter et al. (2013). Watson and Pedatella (2018) derived characteristics of medium-scale F-region plasma irregularities as observed by the COSMIC radio occultation receivers. They analysed 2 to 50 km vertical fluctuations of the observed TEC profiles. The most intense equatorial irregularities are observed around 20:00-24:00 magnetic local time, and correspond to a decrease in the average irregularity scale-size.

Our study is related to the study of Watson and Pedatella (2018) but we are now including an analysis of the change of TEC irregularities before and during a geomagnetic storm. Our study takes advantage of the dense spatio-temporal sampling of ionospheric occultations provided by six LEO satellites. Section 2 describes the GPS radio occultation mission COSMIC and the data analysis for the extraction of ionospheric fluctuations. The results are shown and discussed in section 3. The new result of this work is to study the behaviour of the method presented by Watson and Pedatella (2018) under the presence of ionospheric irregularities during the geomagnetic storm of 15 July 2012.

## 2 Instrument, data and analysis

The joint Taiwan - U.S. Constellation Observing System for Meteorology, Ionosphere, and Climate/Formosa Satellite Mission 3 (COSMIC/FORMOSAT-3, hereafter COSMIC), a constellation of six microsatellites, was launched on 15 April 2006 into a 512-km orbit. After launch the satellites were gradually deployed to their final orbits at 800 km, a process that took about 17 months (Anthes et al., 2008).

The study is based on reprocessed profiles of electron density ($N_e$) and total electron content (TEC) from the COSMIC mission. The TEC profiles describe the total electron content along the ray path from the GPS to the LEO satellite as function of the ray path. The small bending of the ray is neglected in the ionosphere. The analysed data are level1 data (podTec) and level2 data (ionPrf) which were processed by the University Corporation for Atmospheric Research (UCAR) in Boulder (USA). The data are provided in the directory cosmic2013 of the COSMIC Data Analysis and Archive Center (CDAAC). The applied retrieval technique of the $N_e$ profiles is the Abel inversion which assumes local spherical symmetry. The number of electron density profiles is about 1000 per day with a good global coverage. The altitude sampling rate is about 1 km, and the tangent point moves in average 180 km through the ionosphere at altitudes from 400 to 500 km during about 5 minutes. Thus, the profiles are usually not measured above a fixed geographical location. This means that plasma fluctuations in the horizontal, vertical and temporal dimension may contribute to the small-scale fluctuations of an electron density profile or an TEC profile. We assume that the plasma is frozen so that we do not care about temporal fluctuations. Further, we do not try to distinguish between horizontal and vertical fluctuations. Instead, we consider the fluctuation in the s-domain where s is the distance between the bottom tangent point and the tangent point. The bottom tangent point is at the lowest altitude which is recorded for the ionospheric occultation event. The height of the bottom tangent point is usually between 50 and 150 km. In addition, we interpolate the profile to an equally spaced s-grid with a spacing of 1 km. Generally, the tangent point approximately moves along a straight line trajectory in the $F$ region. Hence, the small-scale fluctuations are plasma fluctuations which are projected to the trajectory line of the tangent point of the occultation event. The sounding volume at the tangent point is like a cylinder with a length of about 200km in direction of the GPS-LEO ray, and about 2 km across the ray and about 1 km in altitude. Thus, small-scale fluctuations in ray direction can be smoothed out, occasionally.

We extract the fluctuations in electron density and TEC by means of high pass filtering in the s-domain. In case of TEC, we have to compute the location of the ray tangent point (height, latitude and longitude) by using the coordinates of the GPS and the LEO satellite in the podTec file. The profiles $N_e(s)$ or $\text{TEC}(s)$ are filtered with a digital non-recursive, finite impulse response (FIR) high pass filter performing zero-phase filtering by processing the profiles in forward and reverse directions. A cutoff scale length of 50 km was selected that means that oscillations in electron density with wavelengths less than 50 km are passing the filter. The number of filter coefficients corresponds to three times of a 50 km-interval, and a Hamming window has been selected for the filter. Thus, the high pass filter has a fast response time to vertical changes in the electron density profile. More details about the digital filtering are given by Studer et al. (2012).

The left panel of Figure 1 shows an example of a disturbed electron density profile (blue line) in the southern polar region during the geomagnetic storm of 9 March 2012. In addition, the red line shows the low-pass-filtered profile with scales $> 50$km.

The right panel shows the high-pass-filtered electron density fluctuations $N_{e,1}$ with scales $< 50$km. A similar analysis is performed for the TEC profiles of the same occultation event in Fig. 2. TEC is the total electron content along the horizontal GPS-LEO link where the link is characterized by a certain tangent point height.

For each fluctuation profile (right panel of Fig. 1 or Fig. 2), we compute the mean of the absolute fluctuations within the altitude range 400-500 km which is called $\Delta N_e$ or $\Delta$TEC and which is a measure of the mean amplitude of high frequency fluctuations. The $\Delta N_e$ or $\Delta$TEC-values of the selected profiles are binned into $5° \times 5°$ latitude-longitude grid cells and are averaged by the median function in order to get the global distribution of $F$ region irregularities.

## 3 Results

First at all, we like to compare the global maps for $\Delta N_e$ and $\Delta$TEC. Figure 3a) shows the result of $\Delta N_e$, and Figure 3b) shows the result for $\Delta$TEC both for September 2013 where the COSMIC satellites collected about 30'000 occultation events. Both images have quite similar patterns with enhanced fluctuations in the magnetic polar regions. The coordinates of the geomagnetic and magnetic pole were provided by the World Data Center for Geomagnetism in Kyoto. Generally, the $\Delta$TEC values are a bit enhanced compared to the $\Delta N_e$ values. We suppose that the $\Delta$TEC values are more reliable than the $\Delta N_e$ values since the $\Delta N_e$ values require the Abel inversion and the assumption of local spherical symmetry of the ionosphere. Thus, we provide in the following only the results for the $\Delta$TEC values.

Another question is the influence of the averaging method on the retrieved global map. Figure 4a) shows the $\Delta$TEC map of September 2013 for the case that the arithmetic average is applied to the binned values in the grid cells. On the other hand, Figure 4b) shows the result if the median function is applied to the $\Delta$TEC values. Generally, the arithmetic mean leads to higher values. Especially at the equator, there are some strong fluctuations which may result from the sporadic appearance of equatorial plasma bubbles in the $F_2$-region. However, in case of monthly global maps we prefer the median-function since it reduces the effect of outliers in the data. In case of daily maps, we only have about 1000 occultation events for the globe, and here it seems to be better to apply the arithmetic mean. It is not good to apply the median function if only a few values are present.

Figure 5 shows the dependence of $\Delta$TEC on local time and magnetic latitude during September 2013 obtained by the median function for tangent points between 400 and 500 km altitude. At low latitudes, there is an enhancement of the strength of irregularities after sunset and before midnight. The increase of the strength of the $F_2$ region irregularities is possibly due to spread $F$. The post-sunset rise of the equatorial $F$ layer is regularly seen in ionospheric measurements, and this phenomenon is associated with the occurrence of spread $F$ (Tsunoda, 1981; Tsunoda et al., 2018). One form of spread $F$ are the equatorial plasma bubbles which are upwelling during the post-sunset phase of the equatorial ionosphere (Tsunoda, 1981). At high magnetic latitudes, $\Delta$TEC is enhanced at each local time in Fig. 5.

It is also interesting to investigate the solar cycle effect in the $\Delta$TEC global maps. Figure 6a) shows $\Delta$TEC at solar minimum in September 2008 while Figure 6b) depicts $\Delta$TEC at solar maximum in September 2013. Particularly, the fluctuations are stronger in the magnetic polar regions during solar maximum. A small increase is observed for the equatorial TEC fluctuations

during solar maximum. The observations of a sharp transition between high-latitude irregularities and those outside of the polar region is in agreement with Fejer and Kelley (1980).

A large geomagnetic storm was on 15 July 2012. The geomagnetic index Ap had a value of 78, and a maximum Kp value of 7 was reached. We selected this event since the geomagnetic storm was not preceded by another storm. The storm started after 18:00 UT on 14 July 2012. Figure 7 shows the BX, BY and BZ components of the interplanetary magnetic field (near to the Earth) as provided by the Omniweb data center of the National Aeronautics and Space Administration (NASA). Further the temporal evolution of the Kp index is shown in the bottom panel. For the data analysis, two days-intervals are indicated by the vertical lines for the quiet phase and the storm phase in Fig. 7 . During the storm phase, positive anomalies occur in BX and BY while a negative BZ anomaly is present. That means BZ is southward and the interplanetary magnetic field can reconnect with the magnetospheric field lines which results in a high geomagnetic activity during the storm. According to theory and former observations, the positive deviations of BX and BY during the storm phase shall generate an asymmetric storm response in the Northern and Southern hemisphere. Zakharenkova and Astafyeva (2015) found that geomagnetic activity is larger in the Southern winter hemisphere than in the Northern summer hemisphere. The 15 July 2012 geomagnetic storm was analysed in detail by Wang et al. (2013).

In the following, we average the TEC disturbances over all local times during the quiet phase (12.07.2012 12:00:00 UT to 14.07.2012 12:00:00 UT) and compare this result to the storm phase (14.07.2012 12:00:00 UT to 16.07.2012 12:00:00 UT). Here, we use the arithmetic average since the number of occultation events is not sufficient for the median function. We compare the global maps of $\Delta$TEC (with tangent points at h=400-500km) during the quiet phase (Figure 8a) , and during the geomagnetic storm phase (Figure 8b). The $\Delta$TEC-values of the selected profiles are binned into $10° \times 10°$ latitude-longitude grid cells. The spatial resolution in Figure 8 is selected to be lower than in Figure 6 since the number of occultation events is 1405 for the quiet phase and 1993 for the storm phase. Comparing Figures 8a) and b), it is evident that $\Delta$TEC is increased during the storm phase. Particularly, the Southern magnetic polar region shows a strong increase of the $\Delta$TEC values. There are also patterns of enhanced $\Delta$TEC values at low latitudes during the storm phase.

Figure 9 shows the behaviour of the scintillation index S4 during the quiet phase and the storm phase of the geomagnetic storm of 15 July 2012. S4 is provided by the COSMIC data center (scnLv1 files) and is derived from the amplitude scintillations of the GPS signal at a certain tangent point height. For the global map, the arithmetic means of the S4 values with tangent point heights between 400 and 500 km were taken. Figure 9 shows similar patterns as Fig. 8. During the storm phase enhanced S4 values are found at low latitudes and in the Southern magnetic polar region. Compared to Fig. 8, the S4 map of Fig. 9 has a smaller contrast.

Finally, we like to know how the geomagnetic storm acts on the lower ionosphere where we perform the same analysis but for TEC with ray tangent points from 200 to 300 km altitude. Figure 10 shows the result of $\Delta$TEC for h=200-300 km, and it can be compared to Fig. 8 which showed the results for ray tangent points at h=400-500 km in the upper ionosphere. The number of analysed occultation events is 1176 during the quiet phase and 1603 during the storm phase. The number of occultation events is a bit smaller in Fig. 10 than in Fig. 8 since some occultations did not reach down to 100 km ray tangent point height which we took as a lower limit to ensure a good filtering process of the TEC values with tangent points at h=200-300 km. It

is obvious that the quiet phase is not so quiet in the lower ionosphere where still TEC variations occur in the polar regions and at low latitudes. During the storm phase (Fig. 10b), the disturbed polar regions are extended and the intensity is stronger compared to the quiet phase. In addition the disturbances at low and middle (northern) latitudes are increased during the storm phase. It is obvious that the patterns of enhanced $\Delta$TEC values at low and middle latitudes strongly vary with longitude. In case of the lower ionosphere (Fig. 10) the North-South asymmetry of the storm response is not so pronounced like in the upper ionosphere (Fig. 8).

## 4 Conclusions

The study is based on reprocessed profiles of electron density and total electron content from the COSMIC mission. We applied a special analysis method to extract the spatial fluctuations in electron density and total electron content where we calculate the mean value of the absolute values of the 50 km-high pass filtered fluctuations in the altitude region from 400 to 500 km. The analysis method filtered the irregularities along the path of the tangent point.

The global maps of $\Delta$TEC are quite similar to those of $\Delta N_e$. We find a significant difference if the arithmetic mean or the median is applied to the global map of September 2013. In agreement with numerous ionospheric observations from literature, $\Delta$TEC is enhanced during the post-sunset rise of the equatorial ionosphere in September 2013. The post-sunset rise is associated with spread $F$ and equatorial plasma bubbles (Tsunoda, 1981). At high magnetic latitudes, $\Delta$TEC is enhanced during each hour of the day in September 2013.

The global map of $\Delta$TEC at solar maximum (September 2013) has stronger fluctuations than those at solar minimum (September 2008). We find a new result when we compare the global maps of the quiet phase and the storm phase of the geomagnetic storm of 15 July 2012. It is evident that the TEC fluctuations (ray tangent points at h=400-500km) are increased and extended over the Southern magnetic polar region during the storm phase. This North-South asymmetry is possibly caused by the positive deviations of the BX and BY components of the interplanetary magnetic field. Similar results but less contrast are provided by the global maps of the scintillation index S4.

We find enhanced TEC fluctuations at low latitudes but confined to certain areas. In the lower ionosphere (ray tangent points h=200-300 km), the North-South asymmetry of the geomagnetic storm response is less pronounced than at upper altitudes. The spatio-temporal sampling of the ionosphere by the six LEO satellites of the COSMIC mission is actually not sufficient for a case study of a geomagnetic storm. However, our study gives a first impression on what can be achieved in some future if a larger number (e.g., $> 10$) of LEO satellites with GPS receivers would be launched.

## 5 Code availability

Routines for data analysis and visualization are available upon request by Klemens Hocke.

## 6 Data availability

The level2 data are avaliable at the COSMIC Data Analysis and Archive Center (CDAAC).

*Author contributions.* Klemens Hocke carried out the data analysis. All authors contributed to the interpretation of the data set.

*Competing interests.* The authors declare that they have no competing interests.

5 *Acknowledgements.* We thank the two reviewers for their corrections of the manuscript. We thank the COSMIC Data Analysis and Archive Center (CDAAC). The study was supported by Swiss National Science Foundation under grant number 200021-165516. N. P. was supported by U.S. National Science Foundation grant AGS-1033112. The National Center for Atmospheric Research is sponsored by the National Science Foundation. H. L. acknowledges support from JSPS KAKENHI grants 18H01270, 18H04446, and 17KK0095. G. M. thanks the National Natural Science Foundation of China (NSFC No. 11473045, and 11503040).

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

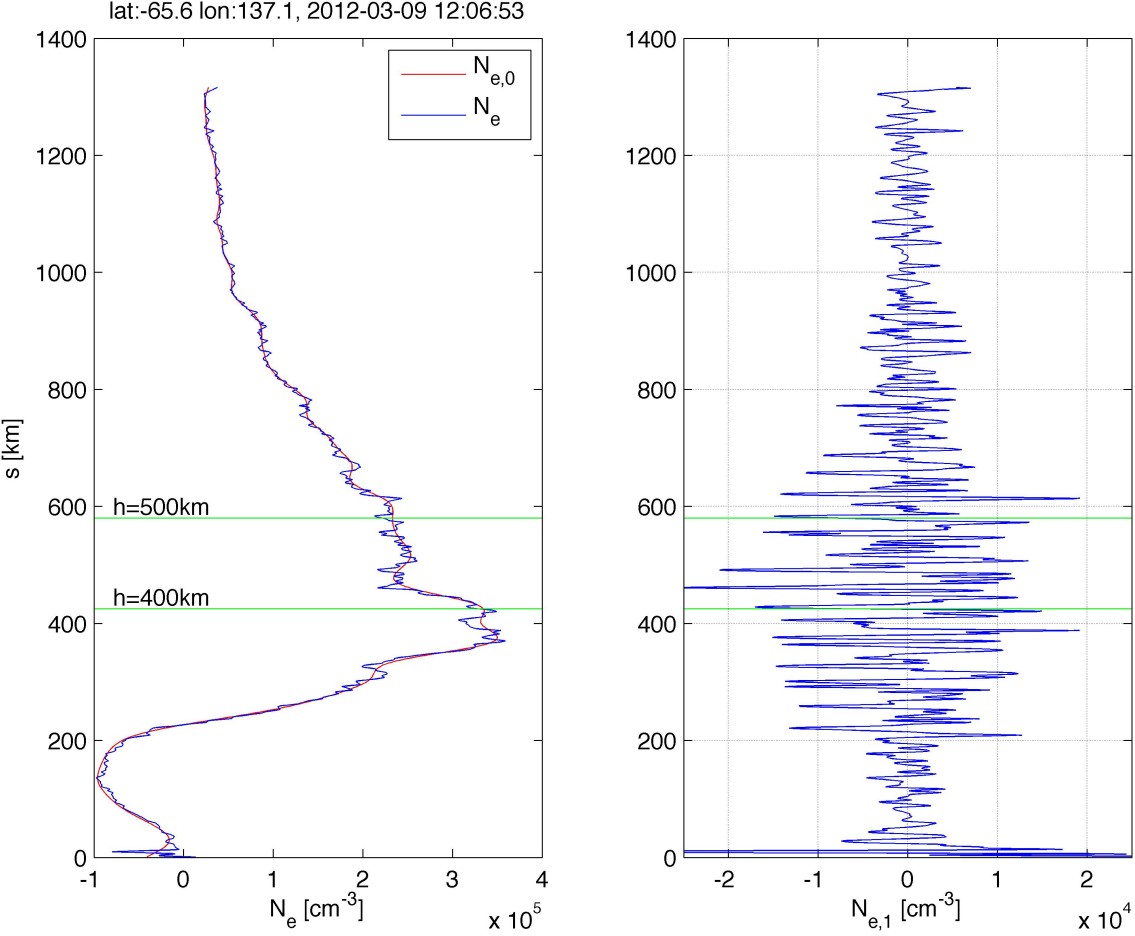

**Figure 1.** Example of a disturbed electron density profile from COSMIC (blue line in the left panel). The red line denotes the 50 km low pass filtered data. The filtering is applied in the s-domain where s is the distance between the bottom tangent point and the tangent point. The right panel shows the electron density fluctuations filtered with the 50 km high pass filter. The study is focused on the altitude region h=400-500 km (with exception of Fig. 10) .

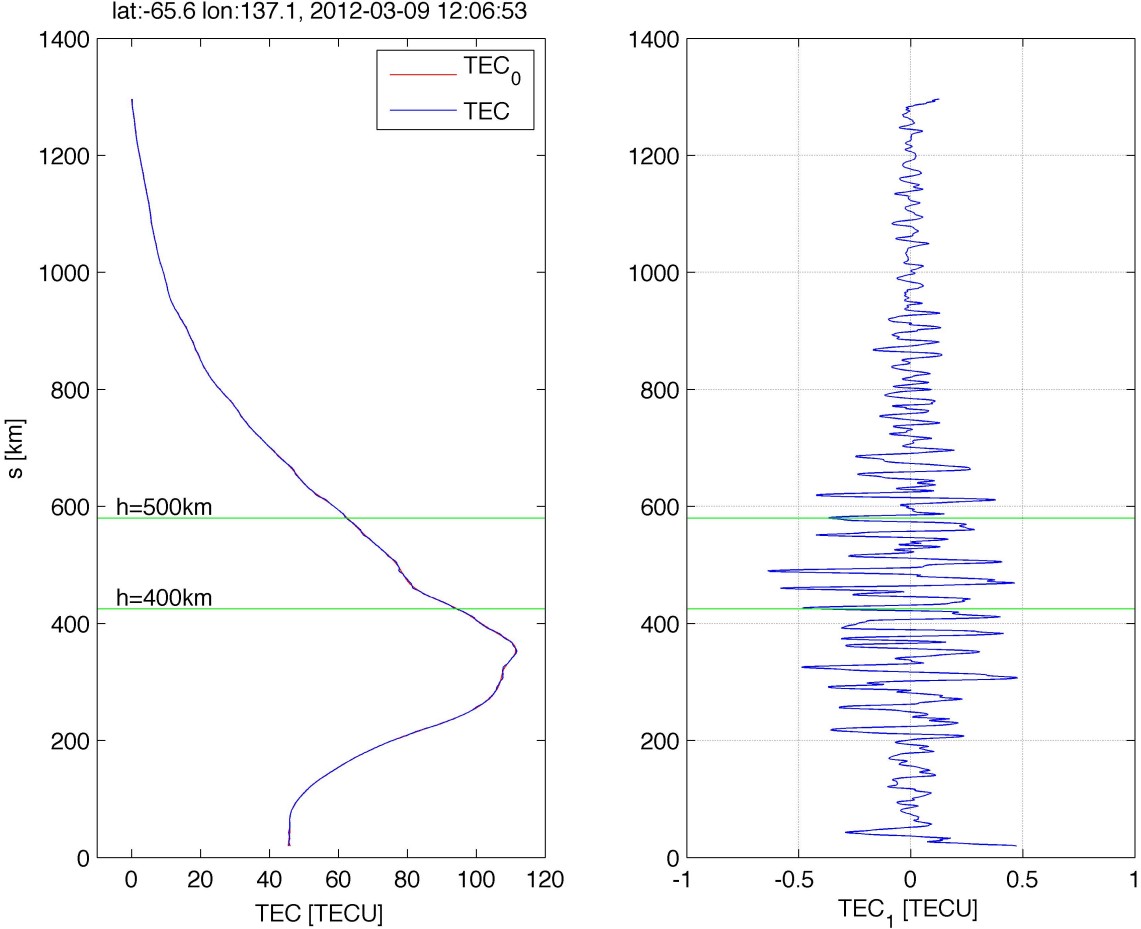

**Figure 2.** Example of disturbed TEC profile from COSMIC (blue line in the left panel). TEC is the total electron content along the GPS-LEO link and measured in TEC units (TECU). The red line denotes the 50 km low pass filtered TEC data. The filtering is applied in the s-domain where s is the distance between the bottom tangent point and the tangent point. The right panel shows the TEC fluctuations filtered with the 50 km high pass filter. The study is focused on the altitude region h=400-500 km (with exception of Fig. 10) .

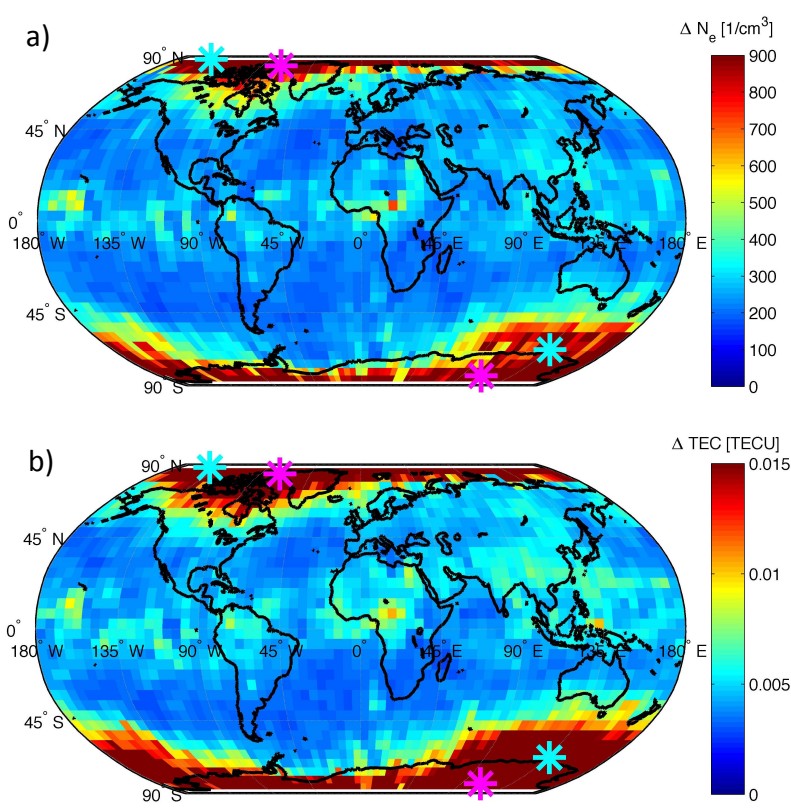

**Figure 3.** a) Global map of $\Delta N_e$ during September 2013 (analysis of ionPrf files of COSMIC). b) Global map of $\Delta$TEC during September 2013 (analysis of podTec files of COSMIC). Both images are derived for fluctuations with scales $< 50$km in the height range 400-500km. The median function is applied to the binned cells ($5° \times 5°$ in latitude and longitude). The geomagnetic (magnetic) poles are indicated by the magenta (cyan) star symbols respectively.

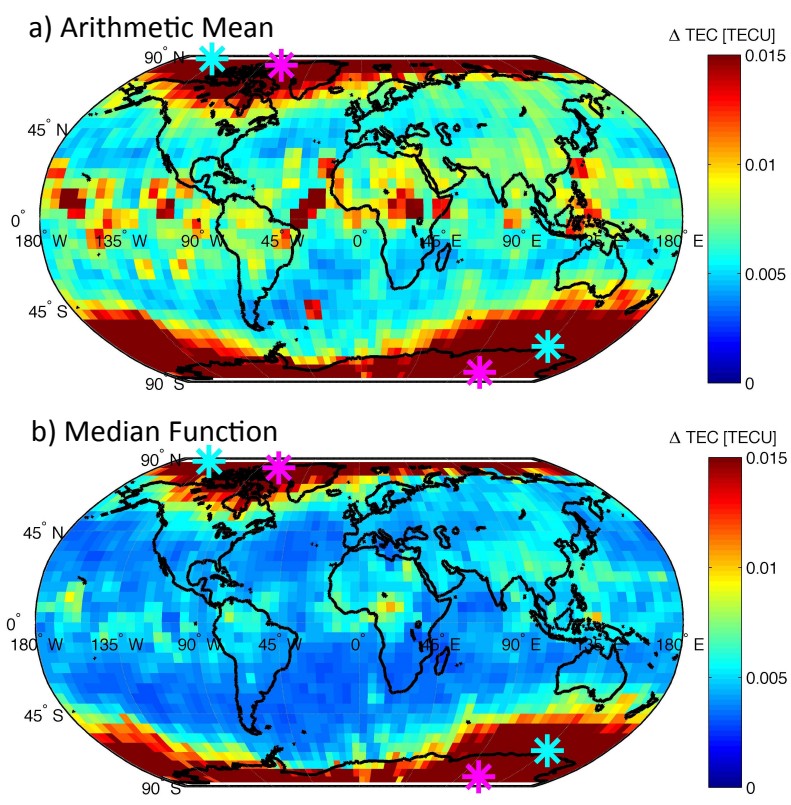

**Figure 4.** a) Global map of $\Delta$TEC during September 2013 obtained by the arithmetic mean of the values in the binned cells. b) Global map of $\Delta$TEC during September 2013 obtained by the median of the values in the binned cells. The geomagnetic (magnetic) poles are indicated by the magenta (cyan) star symbols respectively.

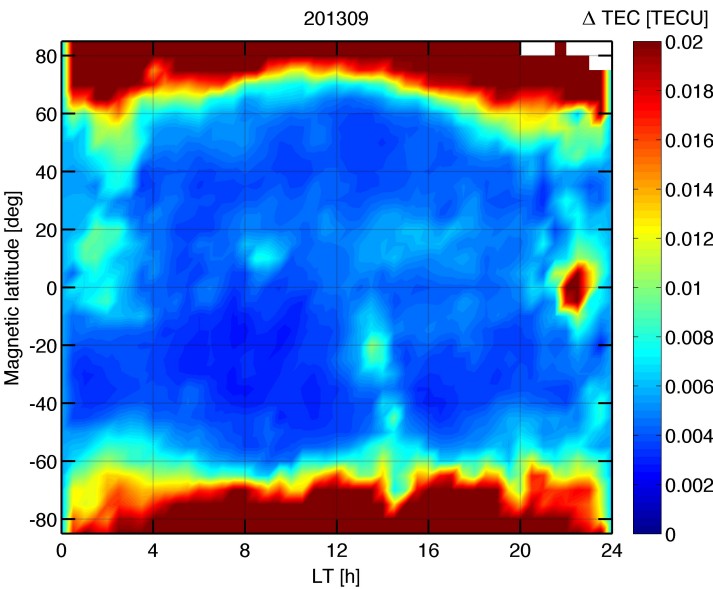

**Figure 5.** Dependence of ΔTEC on local time and magnetic latitude during September 2013 obtained by the median function for tangent points between 400 and 500 km altitude. At low latitudes, there is an enhancement of the strength of irregularities after sunset and before midnight, possibly due to equatorial plasma bubbles.

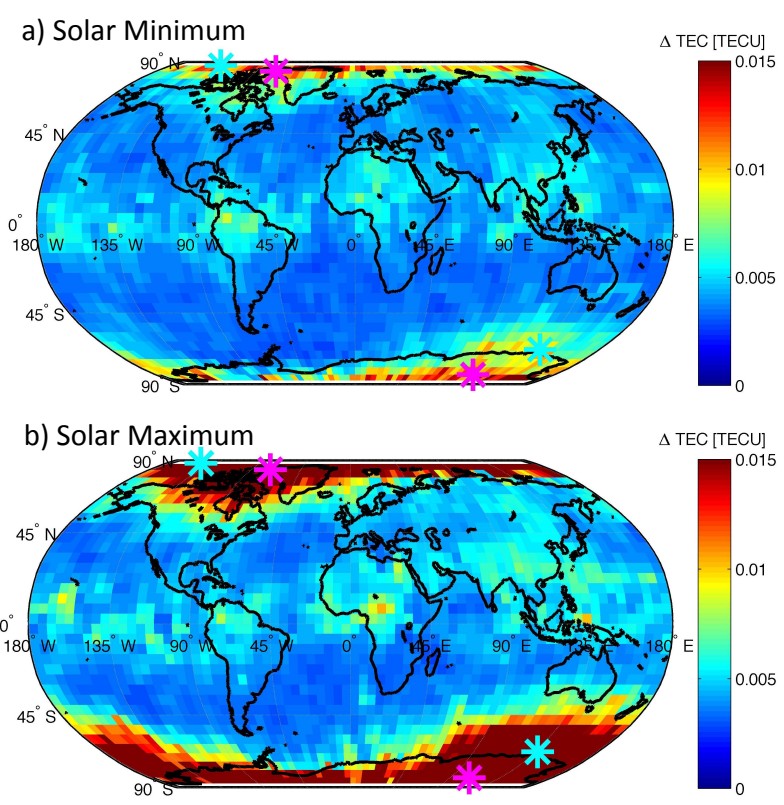

**Figure 6.** a) Global map of ΔTEC during September 2008 (solar minimum). b) Global map of ΔTEC during September 2013 (solar maximum). The geomagnetic (magnetic) poles are indicated by the magenta (cyan) star symbols respectively.

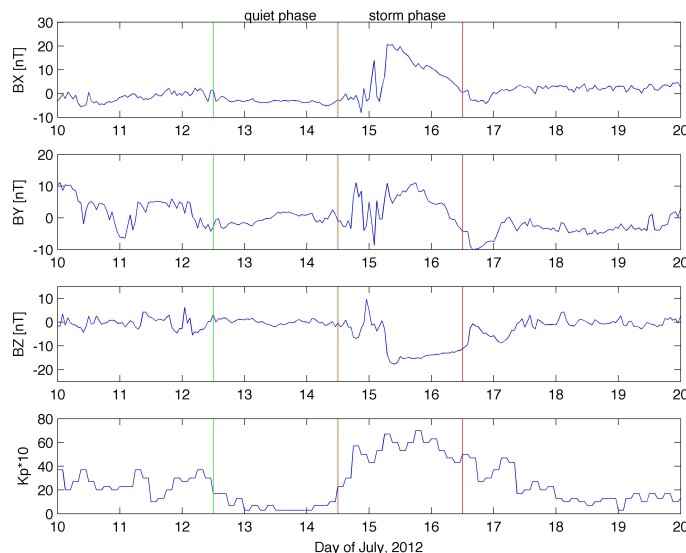

**Figure 7.** BX, BY and BZ of the interplanetary magnetic field (upper panels) and Kp index of the geomagnetic activity (bottom panel). Two days-intervals during the quiet phase before the geomagnetic storm of 15 July 2012 and during the storm phase are indicated by the vertical lines.

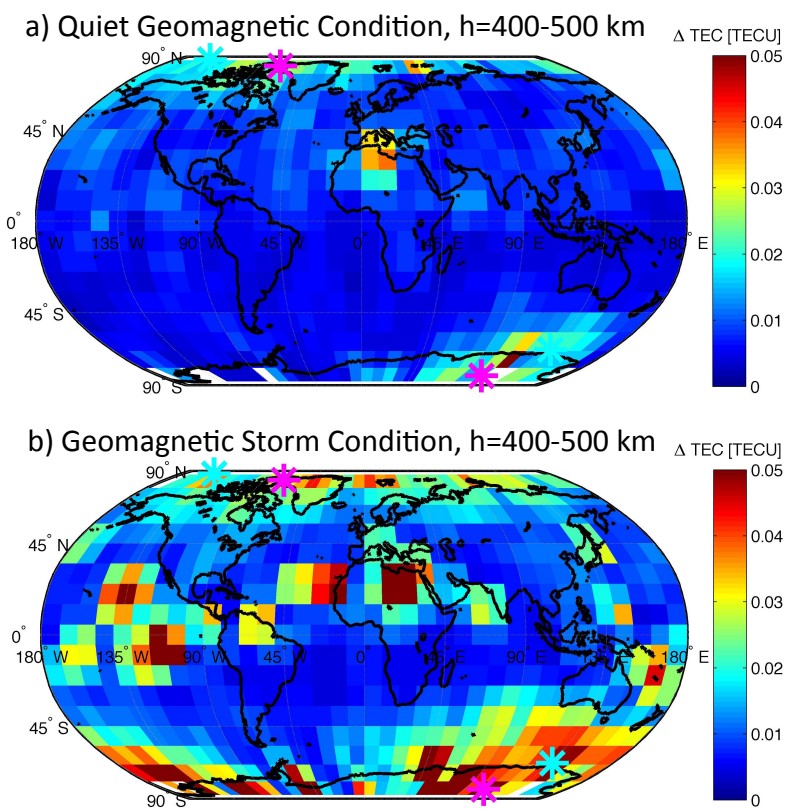

**Figure 8.** a) Global map of $\Delta$TEC (tangent points at h=400-500 km) during the quiet phase, and b) during the storm phase of the geomagnetic storm of 15 July 2012. The arithmetic mean is applied to the values of the binned cells ($10° \times 10°$ in latitude and longitude). The geomagnetic (magnetic) poles are indicated by the magenta (cyan) star symbols respectively.

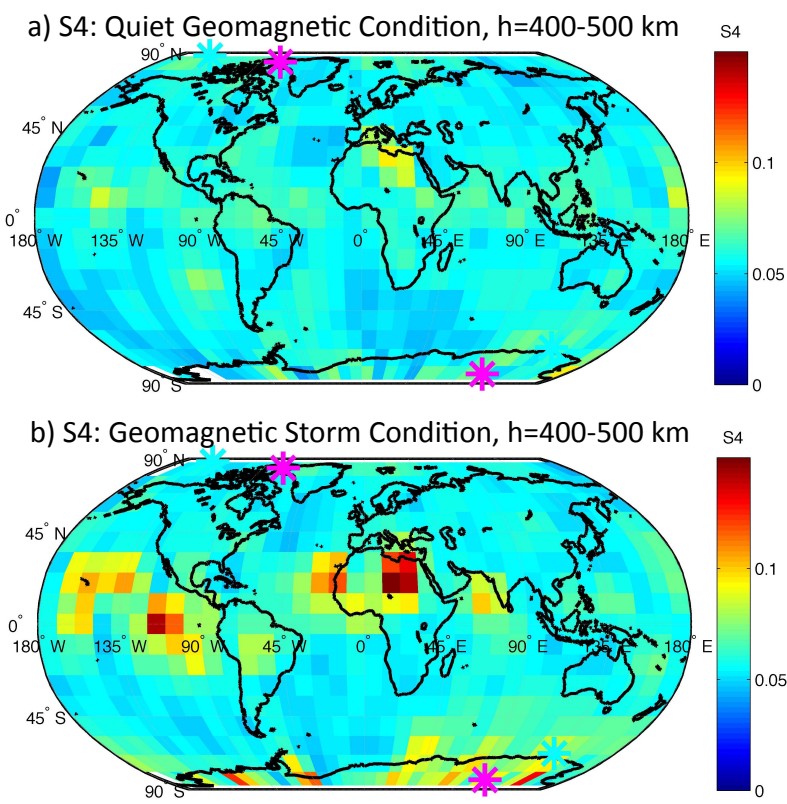

**Figure 9.** a) Global map of the scintillation index S4 (tangent points at h=400-500 km) during the quiet phase, and b) during the storm phase of the geomagnetic storm of 15 July 2012. The arithmetic mean is applied to the values of the binned cells ($10° \times 10°$ in latitude and longitude). The geomagnetic (magnetic) poles are indicated by the magenta (cyan) star symbols respectively.

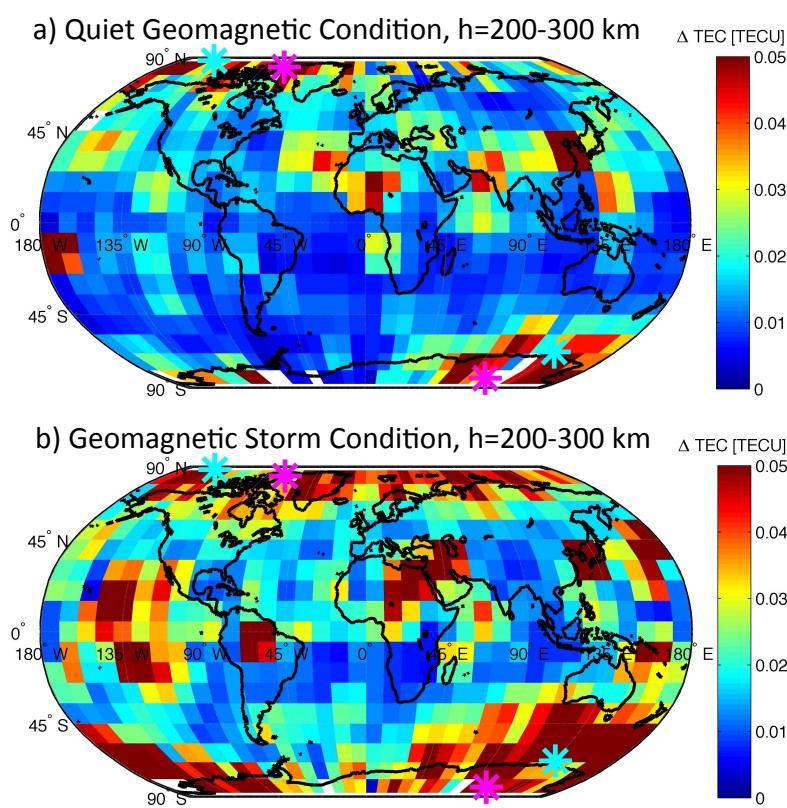

**Figure 10.** a) Global map of $\Delta$TEC (tangent points at h=200-300 km) during the quiet phase, and b) during the storm phase of the geomagnetic storm of 15 July 2012. The arithmetic mean is applied to the values of the binned cells ($10° \times 10°$ in latitude and longitude). The geomagnetic (magnetic) poles are indicated by the magenta (cyan) star symbols respectively.