# Peer review of "Global sounding of F region irregularities by COSMIC during a geomagnetic storm"

_Annales Geophysicae, 2018_

## Referee Comment (RC1) · Anonymous Referee #1 · 12 Dec 2018

The manuscript analyzes global distributions of F-layer ionospheric irregularities by Radio-Occultation techniques during four epochs: solar minimum, solar maximum, quiet phase of a geomagnetic storm and storm phase.

Mapping global characteristics of the irregularities is a difficult task and the authors have shown a possible solution for the global representations.

In the reviewer opinion, the introduction is not addressed with a logical structure. The authors should improve globally the introduction in order to show the reader: 1) an initial contextualization, 2) the related works (state of the art), 3) the objective and 4) the contribution of the new findings (positioning of the manuscript in the state of art). After reading the introduction, it is important for the reader to have a clear context of what is the contribution of the paper in comparison to past works. This is maybe,

from my point of view, the less accurate section of the manuscript. Furthermore, many sentences in the introduction have no connections between each other and a global rephrase of the sentences should improve the manuscript.

The section that describes the used methodology should be improved. After carefully reading the proposed method for obtaining $\Delta$Ne and $\Delta$TEC, the reviewer still found difficult to understand how they were obtained. Including formulations explaining what is $\Delta$Ne and $\Delta$TEC would benefit the manuscript. Indeed, before a proper understanding of what these parameters mean, the reviewer cannot make a fairly evaluation of the analysis.

Considering two occultations with tangent points between specific altitudes (e.g. 400 and 500 km), is $\Delta$TEC the difference between the TEC of one occultation minus the TEC of the other occultation? Please, explain it better. Explain also what is the time resolution between the differences. Also, explain why are you referring this as "TEC profiles".

It is also important to better describe how the high pass filtering in the s-domain was obtained. Additionally, the authors need to be clear with the meaning of the bottom tangent point. The bottom point is located at 400 km in Figure 1? And, in this case of Figure 1, the tangent point is the point located at 500 km?

In the results, it would be important to include distributions not only referred to the longitude but also to the local time. The RO observations do not cover worldwide for every local time. Therefore, sometimes the irregularities seen in a specific location of the maps is not seen in another part because of the different local times. In the way that it is now, each pixel of the global distributions is referred for a distinct instant. The manuscript lacks a proper analysis on this. Even better would be if the authors could plot the maps in terms of magnetic latitude vs local time. Then the authors would be capable of showing a fair global distribution of irregularities.

One last principal question that remains about the manuscript is: Does it is possible

to detect irregularities with such a low spatial resolution of the global representations? The authors said it is possible to detect small-scale fluctuations with spatial scales < 50km with RO. However, the global representation of such information is obtained with a spatial resolution of 5°x5° or 10°x10°. As far as I understood, such maps just give a general information of the number of irregularities in each pixel, but does not describe the irregularities itself. Instead of median, a more informative representation would be the number of times that the gradients of TEC are above some limit (e.g. ∆TEC>0.01 or another value to be defined in the manuscript with a proper reason). Even better would be the percentage of ∆TEC above the defined limit. Then you would have a global representation of irregularities. This because, as far as I understood, the blue up to ~green values are not irregularities, so that, you are not showing maps of irregularities. The way it is now, the irregularities are depending on the spatial resolution of the maps (compare the colorbar of Fig. 4 and 6), which has not a true meaning.

Summing up all these points, the reviewer does not recommend the manuscript for publication before major improvements.

A few other points:

a) In the abstract, COSMIC should read COSMIC/FORMOSAT-3.

b) Section 2 - Include that UCAR has first processed the data level 1 and level 2.

c) pg. 3 - change the word cigar to cylinder.

d) pg. 4 - NASA should read National Aeronautics and Space Administration (NASA)

e) pg. 4 - Citation of Zakharenkova and Astafyeva (2015) is lost in the middle of the text.

f) pg. 4 - "In the following, we average the TEC disturbances over all local times". Did you used the mean (average) or the median?

g) It appears to me that the colorbar of the global Figures (such as Fig. 4) is truncated.

It seems that the maximum value of $\Delta$TEC in the map is higher than the top value of the colorbar. Is that correct? This is just a personal question for you to check. If it is already correct, the authors do not need to include anything in the text or Figures.

---

## Referee Comment (RC2) · Anonymous Referee #2 · 13 Jan 2019

This paper describes the use of COSMIC-GPS observation data (radio occultation grazing/oblique/tangential TEC profiles and Abel-inverted electron density profiles) to monitor the distribution of ionospheric plasma density irregularities around the globe during a geomagnetic storm event on 15 July 2012. High-pass filtered radio occultation TEC and Abel-inverted electron density profiles were used to construct $\Delta$TEC and $\Delta N_e$ parameters, which are akin to the RMS values of the fluctuations. The authors further elaborate that either arithmetic mean or median function can be used for calculating the $\Delta$TEC and $\Delta N_e$ parameters, depending on the specific situation. It is indicated that $\Delta$TEC is preferred since it involves fewer assumptions ($\Delta N_e$ involves uncertain assumptions about spherical symmetry), which I very much agree.

The signal processing/filtering technique and the use of $\Delta$TEC (or $\Delta N_e$) parameters

are clearly useful for monitoring the distribution of ionospheric plasma density irregularities based on the COSMIC-GPS RO measurements. The paper seems to cover all the bases. However, in the present form, the paper appears to be lacking one single unifying emphasis/spearhead that would serve as a strong focal point. Based on my reading of the manuscript, it was not so clear if the desired emphasis of the paper is:

- a demonstration of the usefulness of the dataset and the analysis technique (?), or
- a highlight of the geophysical phenomena consequential to the storm event (?), or
- a broad overview of the expected geospatial distribution of ionospheric irregularities under various condition (?)

I would suggest that the authors emphasize one particular aspect as a focal point, and the discussion of other aspects may revolve around it. I hope this re-organization of abstract/conclusion sections would not be too much to ask.

Furthermore, I would also like to suggest that extra labels are added to some of the figures in order to improve clarity.

Figure 3a: add a label "Arithmetic Mean" on the top of the colormap plot
Figure 3b: add a label "Median Function" on the top of the colormap plot
Figure 4a: add a label "Solar Minimum" on the top of the colormap plot
Figure 4b: add a label "Solar Maximum" on the top of the colormap plot
Figure 6a: add a label "Quiet Geomagnetic Condition, h=400-500 km" on the top of the colormap plot
Figure 6b: add a label "Geomagnetic Storm Condition, h=400-500 km" on the top of the colormap plot
Figure 7a: add a label "Quiet Geomagnetic Condition, h=200-300 km" on the top of the colormap plot
Figure 7b: add a label "Geomagnetic Storm Condition, h=200-300 km" on the top of the colormap plot

I realize that the figure captions listed these information, but including them as labels

in the figure images themselves could potentially be helpful to many readers.

---

## Author Comment (AC1) · 6 Feb 2019

Please find our answers and the marked changes of the revised manuscript in the attached pdf file! Thank you!

Please also note the supplement to this comment:
https://www.ann-geophys-discuss.net/angeo-2018-117/angeo-2018-117-AC1-supplement.pdf

---

## Author Comment (AC2) · 6 Feb 2019

Point-to-point Response:

General remark:
We thank the two referees for their comments and suggestions which have led to an improved, revised manuscript. We reformulated the introduction. We added an image about the dependence of ΔTEC on local time and added a short discussion about spread F during the post-sunset rise of the equatorial ionosphere. In the revised manuscript we also emphasize that the behaviour of F region irregularities during a geomagnetic storm is the main topic of the study.

Referee 1:

In the reviewer opinion, the introduction is not addressed with a logical structure. The authors should improve globally the introduction in order to show the reader: 1) an initial contextualization, 2) the related works (state of the art), 3) the objective and 4) the contribution of the new findings (positioning of the manuscript in the state of art). After reading the introduction, it is important for the reader to have a clear context of what is the contribution of the paper in comparison to past works. This is maybe,
from my point of view, the less accurate section of the manuscript. Furthermore, many sentences in the introduction have no connections between each other and a global rephrase of the sentences should improve the manuscript.

We agree that the introduction was a bit disordered. Now we begin with the main objective (F region irregularities). Then we review studies in the literature about F region irregularities. Then we focus on global maps of F region irregularities and studies related to our study. Finally we give an overview on our article.

The section that describes the used methodology should be improved. After carefully reading the proposed method for obtaining ΔNe and ΔTEC, the reviewer still found dif- ficult to understand how they were obtained. Including formulations explaining what is ΔNe and ΔTEC would benefit the manuscript. Indeed, before a proper understanding of what these parameters mean, the reviewer cannot make a fairly evaluation of the analysis.
Considering two occultations with tangent points between specific altitudes (e.g. 400 and 500 km), is ΔTEC the difference between the TEC of one occultation minus the TEC of the other occultation? Please, explain it better. Explain also what is the time resolution between the differences. Also, explain why are you referring this as "TEC profiles".

ΔTEC is obtained in the same manner as ΔNe where we described ΔNe in detail in Figure 1. Now we added a sentence where we inform that ΔTEC is obtained in the same manner as ΔNe. ΔTEC is not obtained as difference between two occultations. It is the difference between the TEC profile and its smoothed (filtered) TEC profile (as in Figure 1).

It is also important to better describe how the high pass filtering in the s-domain was obtained. Additionally, the authors need to be clear with the meaning of the bottom tangent point. The bottom point is located at 400 km in Figure 1? And, in this case of Figure 1, the tangent point is the point located at 500 km?

We added a sentence that the height of the bottom tangent point is usually between 50 and 150 km.

In the results, it would be important to include distributions not only referred to the longitude but also to the local time. The RO observations do not cover worldwide for every local time. Therefore, sometimes the irregularities seen in a specific location of the maps is not seen in another part because of the different local times. In the way that it is now, each pixel of the global distributions is referred for a distinct instant. The manuscript lacks a proper analysis on this. Even better would be if the authors could plot the maps in terms of magnetic latitude vs local time. Then the authors would be capable of showing a fair global distribution of irregularities.

Yes, we agree, the dependence of ΔTEC on local time is important. The revised manuscript provides a figure for the local time dependence in September 2013 (new figure 4). We find an increase of F rgion irregularities in the post-sunset equatorial ionosphere. At high latitudes the irregularities seem to be independent on local time.

One last principal question that remains about the manuscript is: Does it is possible to detect irregularities with such a low spatial resolution of the global representations? The authors said it is possible to detect small-scale fluctuations with spatial scales < 50km with RO. However, the global representation of such information is obtained with a spatial resolution of 5°x5° or 10°x10°. As far as I understood, such maps just give a general information of the number of irregularities in each pixel, but does not describe the irregularities itself. Instead of median, a more informative representation would be the number of times that the gradients of TEC are above some limit (e.g. ΔTEC>0.01 or another value to be defined in the manuscript with a proper reason). Even better would be the percentage of ΔTEC above the defined limit. Then you would have a global representation of irregularities. This because, as far as I understood, the blue up to ~green values are not irregularities, so that, you are not showing maps of irregularities. The way it is now, the irregularities are depending on the spatial resolution of the maps (compare the colorbar of Fig. 4 and 6), which has not a true meaning.

We analysed the TEC profiles having a high vertical resolution of about 1 km. From the fluctuation profiles, we derived the average global distribution of F region irregularities based on several days or a month of observations. In the past we also tried to analyse TEC gradients and the results were similar. However we think that the analysis as described by Figure 1 is most easy to understand and thus we selected this method.

A few other points:
a) In the abstract, COSMIC should read COSMIC/FORMOSAT-3.

Yes, we changed it.

b) Section 2 - Include that UCAR has first processed the data level 1 and level 2.

Yes, we changed it.

c) pg. 3 - change the word cigar to cylinder.

Yes, we changed it.

d) pg. 4 - NASA should read National Aeronautics and Space Administration (NASA)

Yes, we changed it.

e) pg. 4 - Citation of Zakharenkova and Astafyeva (2015) is lost in the middle of the text.

We started a new sentence with the finding of this study.

f) pg. 4 - "In the following, we average the TEC disturbances over all local times". Did you used the mean (average) or the median?

*It is the mean. We added a new sentence.*

g) It appears to me that the colorbar of the global Figures (such as Fig. 4) is truncated.

*No, the color bar is not truncated.*

*Thank you for your review!*

Referee 2:

Based on my reading of the manuscript, it was not so clear if the desired emphasis of the paper is:
- a demonstration of the usefulness of the dataset and the analysis technique (?), or
- a highlight of the geophysical phenomena consequential to the storm event (?), or
- a broad overview of the expected geospatial distribution of ionospheric irregularities under various condition (?)
I would suggest that the authors emphasize one particular aspect as a focal point, and the discussion of other aspects may revolve around it. I hope this re-organization of abstract/conclusion sections would not be too much to ask.

*We agree. Now we emphasize at various places of the study (e.g., end of introduction) that the focus of our study is the global behaviour of F region irregularities during a geomagnetic storm. This is the new point of the study which was not covered by Watson and Pedatella (2018). We also reformulated the whole introduction section so that the intention of our study becomes clearer.*

Furthermore, I would also like to suggest that extra labels are added to some of the figures in order to improve clarity.
Figure 3a: add a label "Arithmetic Mean" on the top of the colormap plot
Figure 3b: add a label "Median Function" on the top of the colormap plot
Figure 4a: add a label "Solar Minimum" on the top of the colormap plot
Figure 4b: add a label "Solar Maximum" on the top of the colormap plot
Figure 6a: add a label "Quiet Geomagnetic Condition, h=400-500 km" on the top of the colormap plot
Figure 6b: add a label "Geomagnetic Storm Condition, h=400-500 km" on the top of the colormap plot
Figure 7a: add a label "Quiet Geomagnetic Condition, h=200-300 km" on the top of the colormap plot
Figure 7b: add a label "Geomagnetic Storm Condition, h=200-300 km" on the top of the colormap plot

*Good idea! We added the suggested labels in the new figures.*

[revised manuscript text omitted]

---

## Author Response (AR2)

Dear Editor, dear Reviewer,

We thank you  for your comments on the revised manuscript!  Following your advices, we were able to carry out a minor revision which considers all of your comments.

Point-to-point response to Reviewer 1:

1) The main recommendation is to include some sort of small validation of the method during the geomagnetic storm. The calculation of the correlation between S4 indexes and ΔTEC values would be a significant incorporation to the manuscript. S4 indexes can be retrieved by the same RO observations used to derive TEC, which would incorporate a strong analysis. But if this is too much to add, a more feasible validation would be to show a few ionograms with Spread-F during 20:00-24:00 LT at 15 July 2012, confirming the performance of the method.

Good idea! The new version shows the global maps of the S4 index during the quiet phase and the storm phase as provided by the COSMIC data center.  The new Figure 9 (S4) shows similar pattern as the delta TEC values in Figure 8. However, the contrast of the  S4 maps is smaller than those of  the delta TEC maps.

Figure 9 shows the behaviour of the scintillation index S4 during the quiet phase and the storm phase of the geomagnetic storm of 15 July 2012. S4 is provided by the COSMIC data center (scnLv1 files) and is derived from the amplitude scintillations of the GPS signal at a certain tangent point height. For the global map, the arithmetic means of the S4 values with tangent point heights between 400 and 500 km were taken. Figure 9 shows similar patterns as Fig. 8. During the storm phase enhanced S4 values are found at low latitudes and in the Southern magnetic polar region. Compared to Fig. 8, the S4 map of Fig. 9 has a smaller contrast.

2) Pg. 1, line 14 (Abstract): Please, check if it would be better to use "We obtained new results" instead of "We obtain new results".

yes, we changed it to "obtained"

3) In my point of view, the following statement is a bit not connected with the first paragraph of Introduction (pg. 1, line 21):

"GPS radio occultation was already utilized to derive global maps of sporadic E layers around 90-120 km altitude".
Perhaps, it would fit better in the second paragraph.

Good idea! We shifted the sentence to the begin of the second paragraph.

4) Pg. 2, line 1: "One of the main advantages of the GPS radio occultation technique is the high vertical resolution of about 1 km." Advantages in comparison to what? And this sentence is referring to what kind of applications? Ionospheric sounding?

We are now more precise. The best resolution is obtained in the troposphere.  We changed the sentence:

Since the GPS radio occultation technique performs atmospheric limb sounding, the vertical resolution is about 1 km or better in the troposphere.

5) Pg. 2, line 12: "In addition, quasi-dc electric fields". Is DC meaning "direct current"?

Yes

In addition, quasi-direct current electric fields play an essential role in transporting irregular plasma at high-latitudes  \citep{fejer1980}.

6) Pg. 2, line 26: what is MLT?

it's magnetic local time.

The most intense equatorial irregularities are observed around 20:00-24:00 magnetic local time, and correspond to a decrease in the average irregularity scale-size.

7) Pg. 2, line 27: "Our study is related to the study of Watson and Pedatella (2018) but we will add"
In my point of view, it would better to use: "Our study is related to the study of Watson and Pedatella (2018) but we are now including"

okay, thank you

8) Pg. 2, line 30: "The new result and the focus of the study is the behaviour of the ionospheric irregularities during the geomagnetic storm of 15 July 2012". A complete analysis of the ionospheric irregularities during geomagnetic storm would depend on several analysis and several instruments. To keep this sentence, I suggest including more instrumental observation and analysis. Otherwise, I would say that the main goal was not to analyze the irregularities during the storm, but the storm is just a case study to show the performance of the proposed method. Suggestion: "The new result and the focus of this work is to study the behaviour of the method presented by Watson and Pedatella (2018) under the presence of ionospheric irregularities during the geomagnetic storm of 15 July, 2012"

We changed it to:

The new result  of this work is to study  the behaviour of the method presented by  \citet{watson2018}  under the presence of  ionospheric irregularities during the geomagnetic storm of 15 July 2012.

9) pg. 4, line 10: when referring to Fig. 1b, it should be Fig. 1 (right panel). There is no b) in Fig. 1.

okay

For each fluctuation profile (right panel of Fig. \ref{fig1} or  Fig. \ref{fig1a}), we compute the mean of the absolute fluctuations within the altitude range 400-500 km which is called $\Delta N_e$ or $\Delta$TEC and which is a measure of the mean amplitude of high frequency fluctuations.

10) I got a bit confused with the meaning of ΔTEC. As far as I understood, a TEC profile is a plot of TEC versus time (or altitude) along with the tangent point of each occultation. Therefore, a RO-TEC refers to one occulting observation and a RO-TEC profile is all TEC from all observations of the complete occultation of one satellite, which takes some minutes. It took me some time to understand that because, in principle, I was thinking that a TEC profile would be the integral of electron density along with a RO-Ne profile. I would recommend including a plot showing the TEC profile in order to avoid misunderstanding (Like Fig. 1 but for TEC instead of Ne) and explaining it better. Sorry if it obvious for you, but this is an unusual way to see plots of TEC for general readers. Also, it would be interesting to include such kind of plot in order to the reader have a better idea of how is the behavior of a RO-TEC profile.

We agree and we added the new Figure 2 which shows the delta TEC profile.

A similar analysis is performed for the TEC profiles of the same occultation event in Fig. \ref{fig1a}. TEC is the total electron content along the horizontal GPS-LEO link where the link is characterized by a certain tangent point height.

11) I still think that the authors should include an explanation of what is the bottom tangent point. Is this the location of the tangent point obtained in the last (or first) occultation per each profile? I. e., is the bottom tangent point referred to the lowest altitude of each occultation profile?

Yes, it is the lowest altitude of the ionospheric occultation profile

[revised manuscript text omitted]